# Vibrio cholerae Alkalizes Its Environment via Citrate Metabolism to Inhibit Enteric Growth *In Vitro*

Benjamin Kostiuk,[a] Mark E. Becker,[b] Candice N. Churaman,[c] Joshua J. Black,[d] Shelley M. Payne,[e,f] Stefan Pukatzki,[g] Benjamin J. Koestler[c]

[a]Department of Medical Microbiology and Immunology, 6-020 Katz Group Centre, University of Alberta, Edmonton, Alberta, Canada
[b]Department of Cell and Developmental Biology, Northwestern University Feinberg School of Medicine, Chicago, Illinois, USA
[c]Department of Biological Sciences, Western Michigan University, Kalamazoo, Michigan, USA
[d]Department of Molecular Biology and Genetics, Johns Hopkins University School of Medicine, Baltimore, Maryland, USA
[e]Department of Molecular Biosciences, The University of Texas at Austin, Austin, Texas, USA
[f]Institute for Cellular and Molecular Biology, The University of Texas at Austin, Austin, Texas, USA
[g]Department of Biology, The City College of New York, New York, New York, USA

**ABSTRACT** *Vibrio cholerae* is a Gram-negative pathogen, living in constant competition with other bacteria in marine environments and during human infection. One competitive advantage of *V. cholerae* is the ability to metabolize diverse carbon sources, such as chitin and citrate. We observed that when some *V. cholerae* strains were grown on a medium with citrate, the medium's chemical composition turned into a hostile alkaline environment for Gram-negative bacteria, such as *Escherichia coli* and *Shigella flexneri*. We found that although the ability to exclude competing bacteria was not contingent on exogenous citrate, *V. cholerae* C6706 citrate metabolism mutants ΔoadA-1, ΔcitE, and ΔcitF were not able to inhibit *S. flexneri* or *E. coli* growth. Lastly, we demonstrated that while the *V. cholerae* C6706-mediated increased medium pH was necessary for the enteric exclusion phenotype, secondary metabolites, such as bicarbonate (protonated to carbonate in the raised pH) from the metabolism of citrate, enhanced the ability to inhibit the growth of *E. coli*. These data provide a novel example of how *V. cholerae* outcompetes other Gram-negative bacteria.

**IMPORTANCE** *Vibrio cholerae* must compete with other bacteria in order to cause disease. Here, we show that *V. cholerae* creates an alkaline environment, which is able to inhibit the growth of other enteric bacteria. We demonstrate that *V. cholerae* environmental alkalization is linked to the capacity of the bacteria to metabolize citrate. This behavior could potentially contribute to *V. cholerae's* ability to colonize the human intestine.

**KEYWORDS** *Vibrio cholerae*, metabolism, citrate, citrate lyase, oxaloacetate decarboxylase, carbonate, enteric bacteria, pH

For bacteria to thrive in a competitive environment, they must be highly effective in resource acquisition to proliferate their niche (1, 2). Bacteria employ both passive and active forms of competition (2–4). Active processes include the secretion of toxins or the sequestration of resources (1, 3). Passive mechanisms include the secretion of waste products of their secondary metabolic pathways, making the environment hostile for their competitors; such secondary metabolites are not required for the producing organism and are referred to as allelochemicals (5).

*Vibrio cholerae*, the causative agent of the diarrheal disease cholera, is a Gram-negative pathogen that resides primarily in marine reservoirs and causes disease upon human ingestion (6). Because the pathogenic cycle of *V. cholerae* involves transitioning between its natural marine environment and the human host, *V. cholerae* has evolved to be highly

Address correspondence to Benjamin J. Koestler, Benjamin.Koestler@wmich.edu.

The authors declare no conflict of interest.

competitive in both of these environments (6). During these transitions, *V. cholerae* interacts with many different types of microbial organisms, including various eukaryotes and bacteria of the same or other species (7, 8). As a consequence of residing in diverse microbial communities, *V. cholerae* has evolved multiple competitive mechanisms that are effective against other members of its species, other bacterial species, or eukaryotic predators (1, 3, 7).

One such *V. cholerae* survival mechanism is using multiple carbon sources for energy (9). A prominent example is the ability of *V. cholerae* to use chitin as a carbon source, as chitin is abundant in marine environments (10, 11). In addition to chitin, *V. cholerae* can metabolize other carbon sources, such as dietary citrate, that some of its competitors cannot utilize (12). Citrate metabolism is widely conserved among *V. cholerae* strains (13, 14) and contributes to competitiveness in a *V. cholerae* infant mouse model of infection (15). Strains of *V. cholerae* that successfully colonize humans also endure the low pH stress of the stomach and small intestine (9, 16). The capability of *V. cholerae* to thrive in multiple environments has given rise to novel mechanisms of competition. For example, *V. cholerae* actively secretes vibriobactin, and this siderophore sequesters iron to provide this essential nutrient for itself and to prevent other species from using it (1). Other examples include osmotolerance (17), resistance to bile acids (18–20), and biofilm formation.

*V. cholerae* is responsible for seven recorded pandemics. Pandemic strains are divided into two biotypes, namely, the seventh pandemic *V. cholerae* O1 El Tor biotype and the sixth pandemic O1 classical biotype, which evolved as distinct lineages (21–25). Differences in *V. cholerae* metabolism profiles are implicated in significant differences in intraspecies fitness. Notably, some El Tor strains produce 2,3-butanediol when metabolizing glucose; this metabolism creates a more favorable environment for the survival of El Tor strains than classical *V. cholerae* pandemic strains, which generate an unsuitably low pH when grown in the presence of glucose (25, 26). Here, we investigated whether *V. cholerae* C6706, an El Tor strain, uses its ability to metabolize citrate for growth advantages, and we present a model for a competition mechanism in which *V. cholerae* C6706 defines its chemical microenvironment through the metabolism of citrate. Using a simple *in vitro* assay, we show that *V. cholerae* C6706 metabolites create an environment hostile to competing bacteria through the increase in pH and bicarbonate production.

## RESULTS

***V. cholerae* C6706 inhibits the growth of enteric bacteria.** The *V. cholerae* El Tor strain C6706, isolated during the ongoing 7th pandemic (27), suppresses the growth of other microorganisms, such as *Escherichia coli* and *Shigella* spp., in a cross-streaking assay on citrate-containing media when grown for an extended time (54 h) (28). A cross-streaking assay involves growing a streak of *V. cholerae* down the center of an agar plate. Following bacterial growth, bacteria are scraped off. The plate is subjected to chloroform vapor to remove residual bacteria, and a single line of an indicator strain like *E. coli* is streaked perpendicular to the original bacterial growth. The plates are incubated overnight, and the growth of the indicator strain is recorded. A streak of *V. cholerae* C6706 grown on nutrient broth agar supplemented with citrate (CB), but not on nutrient broth agar, resulted in inhibition of the growth of *E. coli* or *Shigella* spp. (29). *E. coli* or *Shigella* spp. did not inhibit *V. cholerae* growth (29). Because *V. cholerae* is no longer present on the agar plate when *E. coli* inhibition occurs, the interpretation was that growth on citrate stimulated *V. cholerae* secretion of an unknown factor.

Two groups had proposed previously that this behavior is competitive where an unidentified bacteriocin-like compound with bactericidal activity against *E. coli* is secreted (28, 30). A third group suggested a metabolic by-product was responsible for this growth inhibition (29). We confirmed this phenotype and found that prolonged growth (48 h) of *V. cholerae* C6706 on lysogeny broth (LB) agar before cross-streaking also resulted in growth inhibition of the closely related enteric bacterium *S. flexneri*, suggesting that the effect was stimulated by, but not dependent on, citrate present in the media for *V. cholerae* C6706

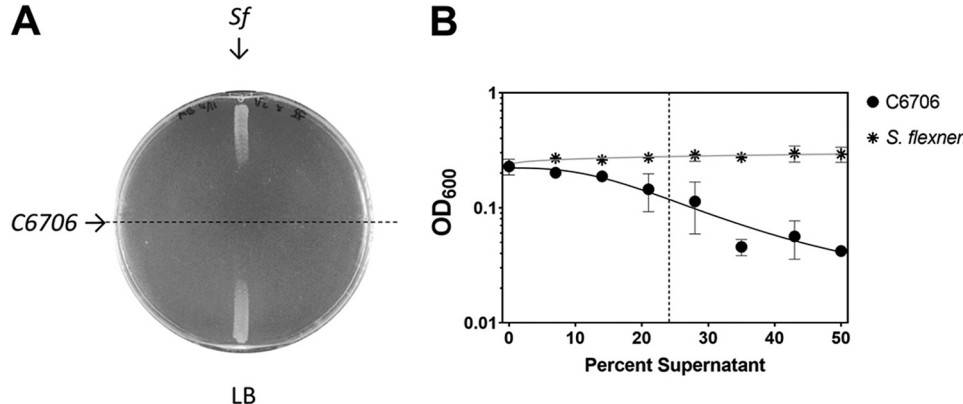

**FIG 1** Extended culture of *V. cholerae* C6706 inhibits enteric growth. (A) A cross-streak assay demonstrates that *V. cholerae* C6706 inhibits the growth of *S. flexneri* CFS100. *V. cholerae* C6706 was grown for 72 h on LB agar. Bacteria were scraped off the plate, followed by a chloroform vapor treatment to kill residual cells. *S. flexneri* CFS100 was then streaked perpendicular to the *V. cholerae* growth and incubated for approximately 12 h. There was a clear zone of inhibition proximal to where the *V. cholerae* was originally located. (B) The supernatant inhibition assay demonstrates that *V. cholerae* cell-free supernatants inhibit *S. flexneri* CFS100 growth. *V. cholerae* C6706 or *S. flexneri* CFS100 was grown in LB for 72 h. Cultures were then centrifuged, supernatants were collected, and supernatants were filtered through a 0.022-$\mu$m PVDF filter. Supernatants were then diluted in LB at different ratios. The ability of *S. flexneri* CFS100 to grow in these supernatants was determined by measuring the $OD_{600}$ after 6 h. *V. cholerae* supernatants inhibited *S. flexneri* growth in a dose-dependent manner, whereas *S. flexneri* supernatants had no inhibitory effects on *S. flexneri* growth. The $IC_{50}$ (indicated by the dotted line) of *V. cholerae* supernatants was calculated to be 24.1%. Each point shows the mean of 3 replicates, and error bars show standard deviation. Trendlines show a nonlinear regression (log inhibitor versus response, four parameters).

(Fig. 1A). We also performed a supernatant inhibition assay, where *V. cholerae* C6706 supernatants were centrifuged and filter sterilized to remove bacteria and debris, and then we measured the ability of the supernatant to prevent *E. coli* or *S. flexneri* growth. This process allowed us to quantitatively compare the inhibition properties of *V. cholerae* C6706 by calculating the half-maximal inhibitory concentration ($IC_{50}$). We found that after growth in LB, *V. cholerae* C6706 cell-free supernatants had dose-dependent inhibitory activity against *S. flexneri*, whereas *S. flexneri* cell-free supernatants did not impede the growth of *S. flexneri* (Fig. 1B).

We sought to determine if the inhibitory effect of *V. cholerae* C6706 supernatants on *S. flexneri* growth in liquid media was bacteriostatic or bactericidal in nature. We grew *S. flexneri* in various concentrations of supernatant derived from the *V. cholerae* C6706 strain and *S. flexneri* and measured growth (optical density at 600 nm [$OD_{600}$]) over time. We then determined the CFS100 growth rate and lag times under each of these conditions. We observed a modest effect on the growth rate of *S. flexneri* when it was grown in various concentrations of supernatant from either strain (Fig. 2A); however, there was a significant dose-dependent increase in the lag time prior to *S. flexneri* growth in *V. cholerae* C6706 supernatant, relative to *S. flexneri* grown in equivalent concentrations of *S. flexneri* supernatant (Fig. 2B), suggesting that this inhibitory effect is largely bacteriostatic in nature.

**Citrate metabolism contributes to *V. cholerae* C6706 inhibition of enteric bacteria.** To determine if the citrate-stimulated competition mechanism of the El Tor strain C6706 is a common trait of *V. cholerae*, we investigated whether other strains share this phenotype. We first determined the ability of 15 *V. cholerae* strains to grow on citrate using Simmons' citrate agar (31). This test relies on an organism to grow using citrate as a sole carbon source. The strains used in this study included both environmental and pandemic-causing strains (32). Surprisingly, we found heterogeneity not only in the ability of various *V. cholerae* strains to create a hostile environment but also on the ability of *V. cholerae* strains to grow using citrate. While all strains were able to grow in LB or CB, we found that 9 of the 15 *V. cholerae* strains could grow on Simmons' citrate agar (Table 1). Of those 9 strains, only five were able to create a hostile environment for *E. coli* in a cross-streak assay (Table 1). *V. cholerae* C6706 was able to grow in

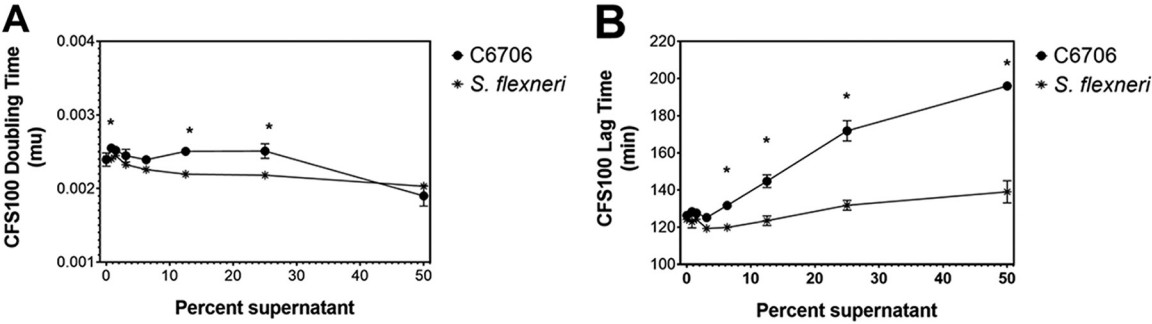

**FIG 2** Effect of *V. cholerae* C6706 supernatants on *S. flexneri* growth. The supernatant inhibition assay was performed using supernatants derived from *V. cholerae* C6706 or *S. flexneri* CFS100, and the growth of *S. flexneri* CFS100 was monitored by OD$_{600}$ measurements. The growth rate and lag time of *S. flexneri* CFS100 grown in different concentrations of supernatant derived from *V. cholerae* C6706 or *S. flexneri* CFS100 was quantified using the R grofit package (68). Each growth curve was replicated in triplicate. (A) We observed a modest increase in doubling time caused by the *V. cholerae* C6706 supernatant, relative to the supernatant derived from *S. flexneri* CFS100. (B) We observed a significant, dose-dependent increase in lag time in *S. flexneri* CFS100 growth caused by *V. cholerae* C6706 supernatants, relative to the *S. flexneri* CFS100 supernatants.

its own conditioned media. Some *V. cholerae* strains (Table 1) grew on citrate provided as the sole carbon source and yet were not able to inhibit the growth of *E. coli*. This finding is consistent with a previous study that noted no correlation between *V. cholerae* citrate metabolism and enteric inhibition from clinical *V. cholerae* isolates (30). Likewise, we did not observe any notable correlation between the source of the *V. cholerae* strains (i.e., pandemic or environmental) and the ability to create a hostile growth environment for *E. coli* (32).

Very few studies have experimentally examined the basis of *V. cholerae* citrate metabolism (15). *V. cholerae* C6706 encodes genes associated with the citric acid cycle (TCA) and also citrate fermentation. In this pathway, a sodium-citrate symporter facilitates citrate uptake and then a citrate lyase (ACLY) cleaves citrate into acetate and oxa-

**TABLE 1** *V. cholerae* strains differ in their ability to metabolize citrate and prevent *E. coli* growth in a cross-streaking assay

| *V. cholerae* producing strain[a] | Results of cross-streak assay | | | |
|---|---|---|---|---|
| | Growth on Simmons' agar | pH of liquid LB medium, 48 h[b] | Growth of *E. coli* MG1655 | Growth of *V. cholerae* C6706 |
| C6706 | + | 9.0 ± 0.2 | − | + |
| C6706 Δ*citE*::Tn | + | n.d. | + | + |
| C6706 Δ*citF*::Tn | + | 8.1 ± 1.1 | + | + |
| N16961 | + | 7.5 ± 1.2 | − | + |
| O395 | + | 6.7 ± 0.1* | − | + |
| DL4211 | + | 9.1 ± 0.2 | − | + |
| 1587 | + | 7.6 ± 1.2 | − | + |
| V52 | + | 7.5 ± 1.2 | + | + |
| MZO-3 | + | 6.7 ± 0.0* | + | + |
| 27-4080 | + | 7.6 ± 1.0 | + | + |
| MAK-757 | + | 7.2 ± 0.1* | + | + |
| AM19226 | − | 7.3 ± 0.0 | + | + |
| V51 | − | 7.6 ± 1.0 | + | + |
| NIH41 | − | 6.9 ± 0.0* | + | + |
| MZ02 | − | 7.1 ± 0.0* | + | + |
| C6709 | − | 7.6 ± 1.0 | + | + |
| CA401 | − | 7.0 ± 0.0* | + | + |

[a]Fifteen *V. cholerae* strains analyzed in this study can be grouped into three distinct groups based on their ability to metabolize citrate as well as their ability to prevent the growth of *E. coli* in a cross-streaking assay on CB. The strains with no shading are able to grow on Simmons' citrate agar and prevent *E. coli* growth. The light-gray-shaded strains are able to grow on Simmons' citrate agar but not prevent the growth of *E. coli*. Finally, the dark-gray-shaded strains are not able to grow on Simmons' citrate agar and subsequently are not able to prevent the growth of *E. coli*. Analysis was replicated at least three times.

[b]For pH values, * indicates statistical significance compared with C6706, as determined by one-way ANOVA with Dunnett's posttest ($P < 0.05$); one outlier was identified and removed using the ROUT test (C6706, pH 6.8). n.d., not determined.

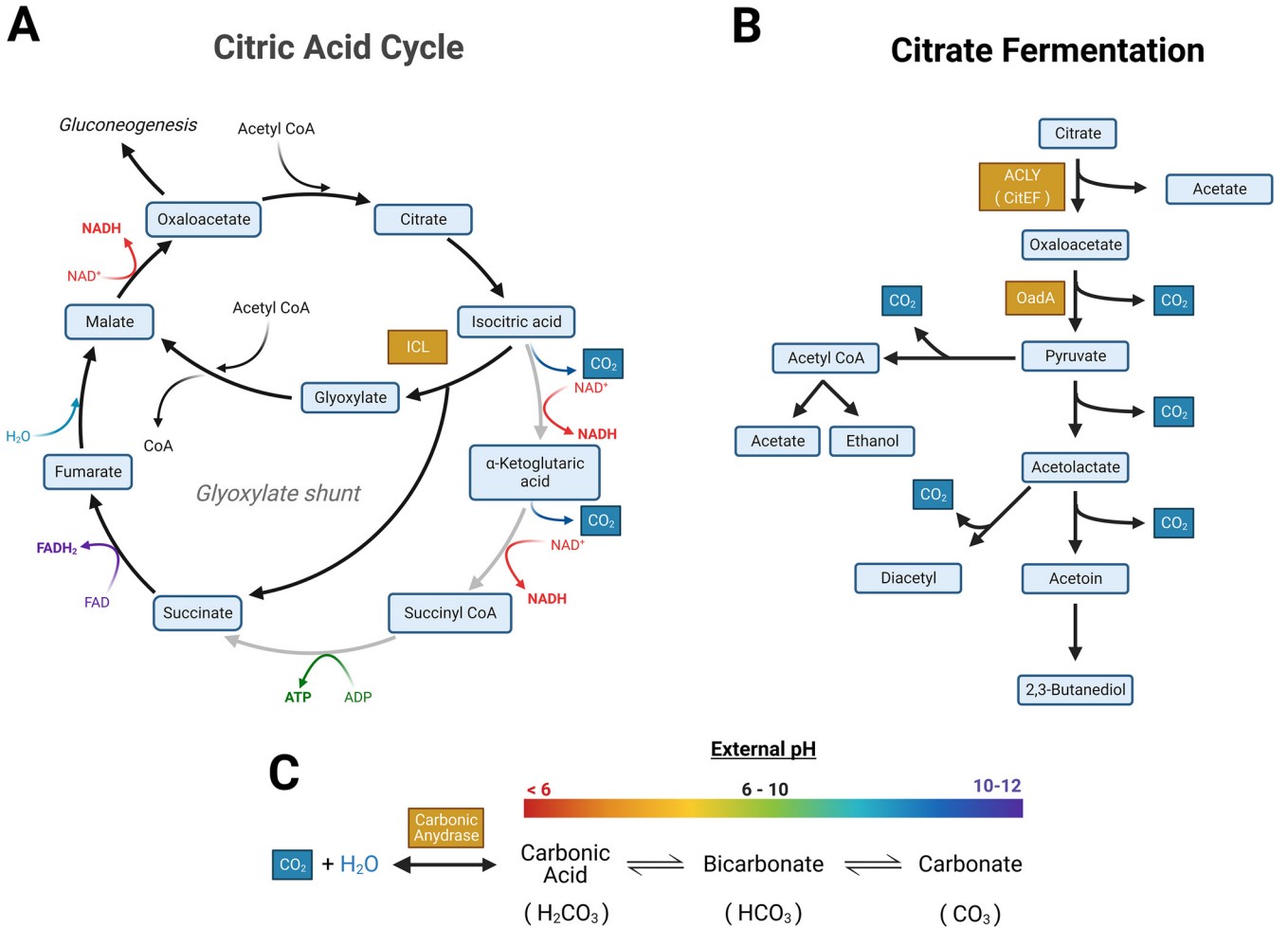

**FIG 3** Conceptual model summarizing that the metabolism of citrate under basic conditions favors the formation of carbonate. (A) Diagram of the citric acid cycle in *V. cholerae*, highlighting the glyoxylate shunt. (B) Abridged metabolic flow diagram showing the fermentation of citrate to 2,3-butanediol or acetyl-CoA, producing $CO_2$ as a by-product (in blue). (C) $CO_2$ is converted to carbonic acid through the enzyme carbonic anhydrase. An equilibrium exists between carbonic acid, bicarbonate, and carbonate. More basic conditions favor the production of the negatively charged carbonate. Enzymes of interest in this study are highlighted in orange. Other enzymes and other cofactors have been omitted to emphasize the production of $CO_2$. Created with BioRender.com.

loacetate, of which oxaloacetate is converted to $CO_2$ and pyruvate by oxaloacetate decarboxylase (33–36) (Fig. 3). To determine if citrate fermentation contributes to enteric growth inhibition, we tested two *V. cholerae* C6706 transposon mutants with disrupted genes encoding ACLY, namely, Δ*citE*::Tn and Δ*citF*::Tn, to determine the importance of the citrate metabolism pathway on the *V. cholerae* C6706 competition phenotype (37). Although both of these mutants still displayed growth on Simmons' citrate agar, they did not prevent *E. coli* growth (Table 1). Similarly, a mutation in *oadA-1*, coding for oxaloacetate decarboxylase, which blocks the conversion of oxaloacetate to pyruvate in citrate metabolism (34), also eliminated the inhibitory activity produced by *V. cholerae* C6706 in a supernatant inhibition assay (Fig. 4A). We concluded that the inhibitory activity of *V. cholerae* C6706 on *S. flexneri* and *E. coli* is dependent on *V. cholerae* C6706 citrate fermentation.

**_V. cholerae_ C6706 raises the pH of media during growth.** We sought to characterize the nature of the *V. cholerae* C6706 inhibition of *S. flexneri*. The secretion of inhibitory factors is a bacterial behavior that often conveys a communal fitness advantage (38, 39); therefore, we hypothesized that the *V. cholerae* inhibitory mechanism was regulated by quorum sensing. However, we found that there was no significant difference in the $IC_{50}$ of wild-type (W.T.) *V. cholerae* C6706 supernatants (Fig. 4B, 29.3, dashed line) compared with that of Δ*hapR* (Fig. 4B, 24.9 dotted line) and Δ*luxO* (Fig. 4B, 23.3, dotted line)

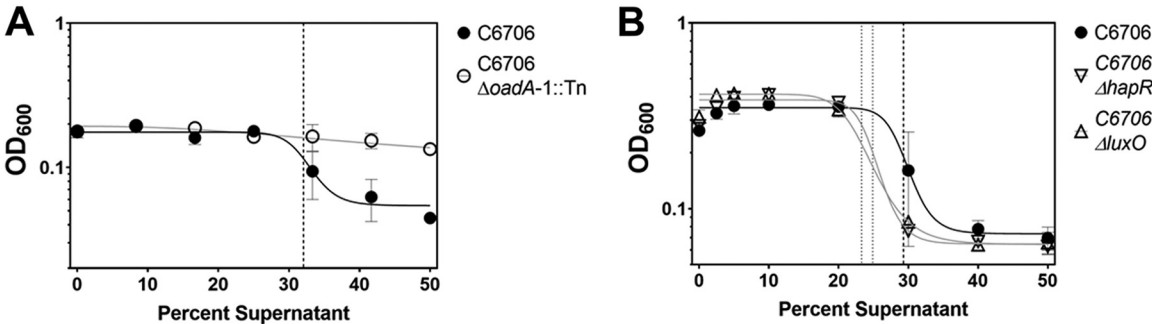

**FIG 4** Quantification of the *V. cholerae* supernatant inhibitory activity of *S. flexneri* using a cell-free assay. (A) Disruption of a *V. cholerae* oxaloacetate decarboxylase (*oadA-1*) eliminates supernatant inhibitory activity. Supernatants derived from the growth of the *V. cholerae* Δ*oadA*-1::Tn mutant were unable to inhibit *S. flexneri* CFS100 growth, compared with the W.T. strain. The IC$_{50}$ of the W.T. strain is shown by the dashed line. (B) Disruption of *V. cholerae* quorum sensing master regulators does not significantly alter supernatant inhibition activity. Supernatants derived from the growth of *V. cholerae* Δ*hapR*::Tn or Δ*luxO*::Tn mutants were able to inhibit the growth of *S. flexneri* CFS100 similar to the *V. cholerae* C6706 W.T. strain. Each point is the mean of 3 replicates, and error bars show standard deviation. Trendlines show a nonlinear regression (log inhibitor versus response, 4 parameter). IC$_{50}$ values are shown by the dashed (W.T.) and dotted (Δ*hapR* and Δ*luxO*) lines.

quorum-sensing mutant strains in our supernatant inhibition assay (Fig. 4B). We also investigated the hypothesis that the *V. cholerae* C6706 secreted factor is a protein (28, 30). If the *V. cholerae* C6706 secreted factor was a protein, it might be sensitive to heat; however, *V. cholerae* C6706 cell-free supernatants that were incubated at 60°C for 60 min still retained their ability to inhibit *S. flexneri* growth (Table 2). Furthermore, proteinase K treatment of *V. cholerae* C6706 cell-free supernatants also was not able to significantly alleviate *S. flexneri* growth inhibition, as well as filtration through a 1-kDa filter (Table 2). These data together do not support the hypothesis that the *V. cholerae* C6706 secreted inhibitory factor is a protein.

We next investigated the hypothesis that the *V. cholerae* C6706 secreted inhibitory factor is a metabolic by-product (29). We performed a methanol extraction to isolate metabolites from *V. cholerae* C6706 cell-free supernatants. Methanol was added to *V. cholerae* C6706 cell-free supernatants at a ratio of 6:1 to precipitate protein. The liquid fraction was collected and evaporated by vacuum centrifugation; the remaining content was resuspended in saline. We found that metabolites extracted from *V. cholerae* C6706 cell-free supernatants did not display any inhibitory activity toward *S. flexneri* (Table 2).

As part of this experiment, we measured the pH of the supernatants before and after extraction and measured the pH of metabolites resuspended in saline after methanol extraction. Prior to culture, the pH of the LB used to grow *V. cholerae* C6706 and *S. flexneri* was approximately 7.0; however, the pH of *V. cholerae* C6706 cell-free supernatants after 48 h of culture was approximately 9.5, consistent with prior studies (29). After methanol extraction, the pH of *V. cholerae* C6706 cell-free supernatant metabolites resuspended in saline was approximately 7.0. We also measured the pH of CB before and after *V. cholerae* C6706 growth. Like LB, the pH of liquid CB was approxi-

**TABLE 2** Supernatant treatment effects on the ability to inhibit *S. flexneri* CFS100 growth[a]

| Supernatant producing strain | Supernatant inhibition assay results | | | | |
|---|---|---|---|---|---|
| | No treatment | Heat | Proteinase K | Filtration | Methanol extraction |
| *V. cholerae* C6706 | + | + | + | + | − |
| *S. flexneri* CFS100 | − | − | − | − | − |

[a]*V. cholerae* C6706 or *S. flexneri* CFS100 were grown for 72 h in LB, cells were removed, and then cell-free supernatants were treated and used in a supernatant inhibition assay to inhibit the growth of *S. flexneri* C6706. Treatments included heat treatment (60°C for 1 h), treatment with proteinase K, filtration through a 1-kDa filter, or methanol phase separation followed by reconstitution in saline. +, indicates supernatants were able to inhibit the growth of *S. flexneri* CFS100; −, indicates that supernatants were not able to inhibit the growth of *S. flexneri* CFS100.

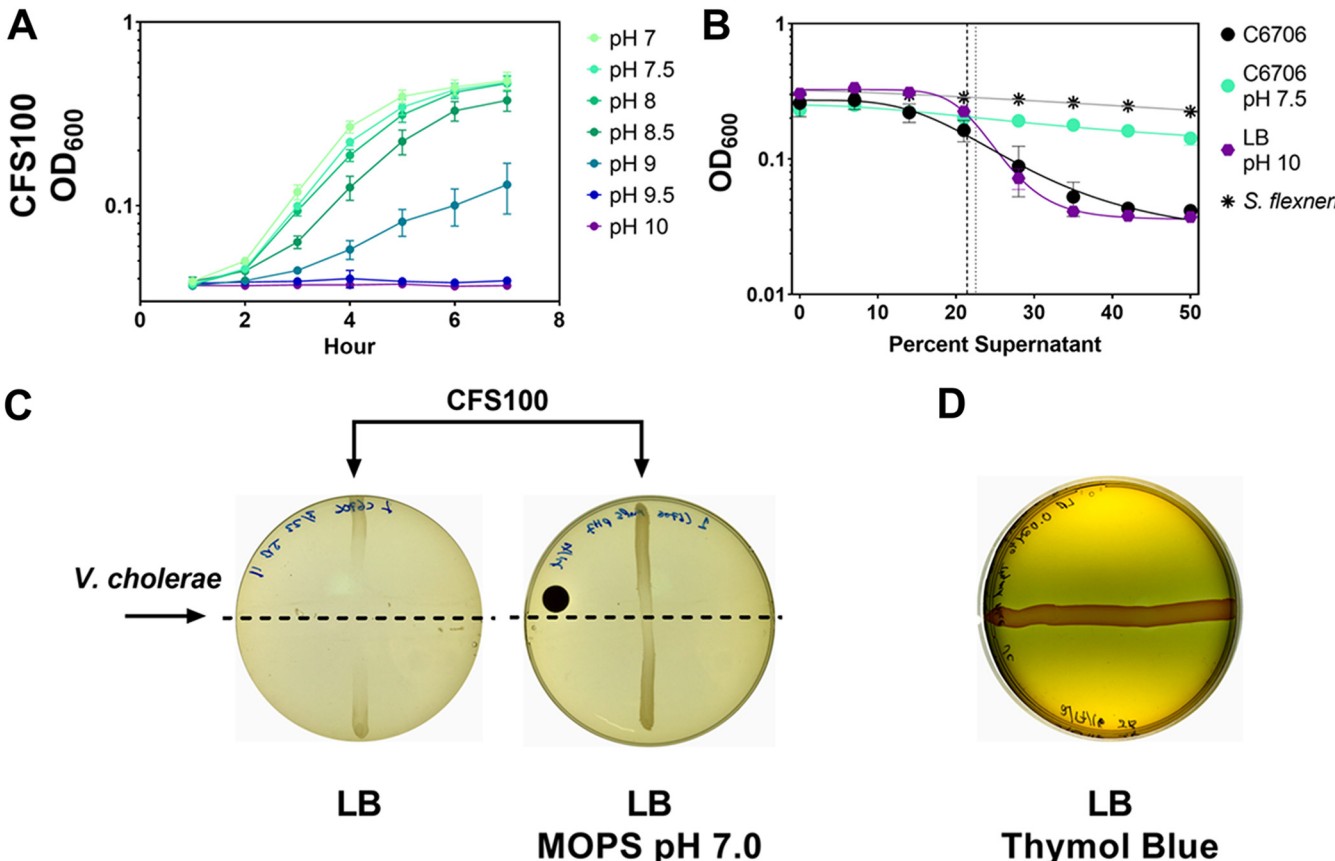

**FIG 5** Media alkalization is necessary for *V. cholerae* supernatant inhibitory activity. (A) Alkaline pH inhibits *S. flexneri* CFS100 growth. LB was adjusted to different pHs using sodium hydroxide, and *S. flexneri* growth was quantified over time by measuring $OD_{600}$. When the media pH exceeded 9.0, significant *S. flexneri* growth inhibition occurred. (B) Changes in pH correspond to *V. cholerae* supernatant inhibitory activity. The supernatant inhibition assay was used to quantify the inhibitory activity of *V. cholerae* supernatants with adjusted pH. When the pH of *V. cholerae* was lowered to 7.5, no *S. flexneri* growth inhibition was observed; in contrast, when the pH of LB was raised to 10.0, the pattern of growth inhibition was similar to that of *V. cholerae* supernatants. (C) Buffering LB agar plates reduces *V. cholerae* supernatant inhibitory activity. LB agar was prepared with and without MOPS buffer (pH 7.0; 50 mM). When the cross-streak assay was performed, there was a minimal zone of inhibition observed in the MOPS plate, compared with the LB agar plate. (D) Thymol blue was supplemented to LB agar to visualize pH changes after *V. cholerae* growth. After 24 h, a blue coloration change is observed surrounding the *V. cholerae* streak corresponding with the zone of inhibition we observe, indicating an increase in medium pH.

mately 7.0 before inoculation; following 48-h growth of C6706 in CB, the pH rose by approximately two logs to 9.0. To determine if other *V. cholerae* strains raise medium pH similar to the C6706 strain, we grew each of our *V. cholerae* strains in LB for 48 h and then quantified the pH of cell-free supernatants. Only the *V. cholerae* C6706 and DL4211 strains demonstrated a pH of >9; notably, there was variation in the medium pH of several *V. cholerae* strains after 48 h of growth (Table 1).

**_V. cholerae_ C6706 alkalization of media inhibits _S. flexneri_ growth.** *V. cholerae* can grow in pHs between 6.5 and 9, with an optimal growth pH of 8 (40, 41); *E. coli* and *S. flexneri* do not grow when the external pH is higher than 9.0 (42). To confirm that *S. flexneri* is also sensitive to basic conditions, we grew *S. flexneri* in LB adjusted to different pHs. We observed a modest growth defect when the pH of LB was brought to 9.0, and we observed no growth when the pH was raised to 9.5 and 10.0 (Fig. 5A). Because the pH of *V. cholerae* C6706 supernatants was higher than 9.0 after 72 h, we hypothesized that alkaline pH contributed to the inhibition of *S. flexneri* growth. We tested this hypothesis by adjusting the pH of *V. cholerae* C6706 cell-free supernatants. When the pH of *V. cholerae* C6706 cell-free supernatants was adjusted to 7.5 using HCl, we found that *V. cholerae* C6706 cell-free supernatants could not inhibit *S. flexneri* growth in a liquid inhibition assay (Fig. 5B).

Furthermore, when the pH of LB was raised to 10.0, it had the same inhibitory effect toward *S. flexneri* as *V. cholerae* C6706 supernatants (Fig. 5B). To determine if this same

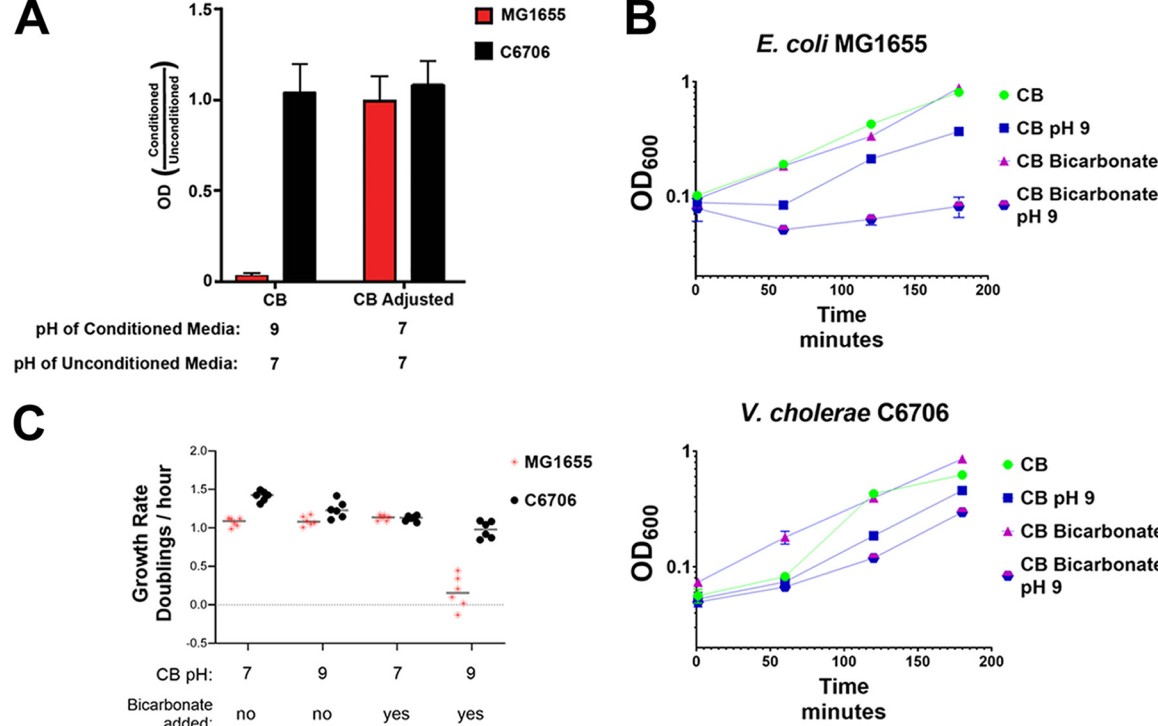

**FIG 6** The growth inhibition of *E. coli* can be replicated in a *Vibrio*-free system using a high pH and bicarbonate. (A) Conditioned CB medium by growth of *V. cholerae* prevents the growth of *E. coli* in a pH-dependent manner. Shown is the optical density of indicated bacterial culture after 4 h of growth in different conditioned media compared with that of unconditioned media. *V. cholerae* C6706 was grown in CB media for 48 h. The pH cell-free supernatant was then either adjusted to 7.0 or left as indicated. The relative ability of both *E. coli* and *V. cholerae* to grow in this conditioned media was determined by the optical density after 4 h compared with the optical density in unconditioned media. (B) Preventing the growth of *E. coli* can be mimicked with pH and bicarbonate. CB medium is either supplemented with bicarbonate or not and either left at pH 7 or artificially raised to a pH of 9. Top, shows the growth of *E. coli* MG1655 in different media conditions; bottom, shows the growth of *V. cholerae* C6706. (C) Growth rates of both *E. coli* and *V. cholerae* were determined in these four media by taking the slope of the linear portion of the growth curve. The *y* axis represents the growth rate of *E. coli* divided by the growth rate of *V. cholerae* under the conditions indicated on the *x* axis.

effect occurred in our cross-streaking assay, we grew *V. cholerae* C6706 on LB agar buffered with morpholinepropanesulfonic acid (MOPS; 50 mM) at pH 7.0. When we performed our cross-streaking assay using this buffered medium, we observed little to no inhibition of *S. flexneri* growth (Fig. 5C). We also visualized *V. cholerae* C6706 alkalization of LB agar by adding thymol blue, a pH indicator that transitions from yellow to blue between pH 8.0 and 9.2. After 24 h, we observed a blue zone surrounding *V. cholerae* C6706 growth, which corresponded to the change in pH we observed in liquid medium (Table 1) and the *S. flexneri* zone of inhibition we observed (Fig. 5D).

Similar to growth in LB, when *V. cholerae* C6706 was grown in CB, we found that *V. cholerae* C6706 cell-free supernatants did not inhibit *E. coli* growth after the pH was readjusted to ~7.0 with HCl or if the *V. cholerae* C6706 growth medium was buffered to prevent the increase in pH (Fig. 6A). Carbonate is a by-product of citrate metabolism under basic conditions and is known to prevent the growth of *E. coli* (43), and thus, we hypothesized that *V. cholerae* C6706 metabolism of citrate generates carbonates and basic conditions responsible for deprotonating bicarbonate to carbonate (Fig. 3). To test the hypothesis, we examined *E. coli* and *V. cholerae* C6706 growth in a combination of sodium bicarbonate and a basic pH, mimicking the conditions we anticipate are caused by *V. cholerae* C6706 grown in citrate-containing media. We found that these conditions restricted the growth of *E. coli* more profoundly than basic pH or bicarbonate alone (Fig. 6B). *V. cholerae* C6706 grew under all conditions, and even though *V. cholerae* C6706 grew more slowly in the combination of high pH and bicarbonate (Fig. 6B), it still reached the mid-logarithmic phase of growth. The supplementation of

bicarbonate to media at a pH of 9.0 reduced the growth rate of *E. coli* but not that of *V. cholerae* (Fig. 6C).

## DISCUSSION

It was first reported some 50 years ago that some *V. cholerae* strains inhibit the growth of enteric bacteria, but the mechanism of this inhibition was not determined (28–30). While more recent studies have illustrated how *V. cholerae* kills other bacteria using a type VI secretion system (T6SS) (3), the mechanism first described by Chakrabarty et al. (28) is a contact-independent mechanism. Here, we investigated the hypotheses originally posited by these groups regarding the *V. cholerae* mechanism of contact-independent inhibition of enteric growth *in vitro* (28–30). Our findings were consistent with the findings of Bhaskaran et al. (29), who proposed that secreted carbonates raised media pH and caused enteric growth inhibition. We provide additional support of this hypothesis by demonstrating that *V. cholerae* C6706 mutants defective in the conversion of citrate to oxaloacetate (*citE* and *citF*) and the conversion of oxaloacetate to pyruvate (*oadA-1*) were not able to inhibit enteric growth (Table 1; Fig. 4A). There are limitations to this study that are worth noting. These experiments are artificial in nature, and they show only the effects of *V. cholerae* secreted products after extended growth. All experiments were performed in lab medium, which is very different from a marine or host intestinal environment, and so we cannot ascertain the significance of enteric inhibition during host colonization. We cannot rule out the possibility that *V. cholerae* C6706 produces a bacteriocin-like protein to inhibit the growth of enteric bacteria; however, our data suggest that such a protein would require an alkaline environment to function, as buffering media pH or readjusting *V. cholerae* C6706 cell-free supernatants to a neutral pH abolishes inhibitory activity (Fig. 5 and 6).

Carbohydrate metabolism is an essential aspect of *V. cholerae* pathogenesis. Previous studies have shown that *V. cholerae* relies on standard carbon metabolism pathways coupled with oxygen respiration in the host, including the Embden-Meyerhof-Parnas (glycolysis) and Entner-Doudoroff pathways (44–46). Recent studies demonstrate that pyruvate dehydrogenase and pyruvate formate lyase, enzymes that facilitate the transition to TCA or fermentation by converting pyruvate to acetyl-coenzyme A (CoA), are essential for *V. cholerae* pathogenesis in an infant mouse model (47) and that citrate metabolism intersects this metabolic node. The citrate metabolic axis in particular is a defining aspect of *V. cholerae*, which is commonly used to differentiate *V. cholerae* from other bacteria (13). Citrate metabolism genes are highly conserved in *V. cholerae* (14), and citrate fermentation promotes *V. cholerae* pathogenesis in an infant mouse model (15).

Notably, *V. cholerae* citrate fermentation, along with lactate and acetate metabolism, produce carbonates as a by-product (29). *V. cholerae* C6706 also encodes at least three putative carbonic anhydrases (VC0586, VCA0274, and VC0058) that potentially contribute to the accumulation of environmental carbonates (48, 49). Carbonates incorporate free H+ ions to produce $CO_2$, which raises pH. Additionally, OadA-1 is a decarboxylase that consumes a proton in the conversion of oxaloacetate to pyruvate, which could also contribute to environmental alkalization (34). The human host also secretes bicarbonate into the intestine (50), and bicarbonate serves as an important signal in *V. cholerae* pathogenesis. Bicarbonate is a critical component of AKI conditions to induce cholera toxin production *in vitro* (51). It has also had more recently been shown that bicarbonate regulates the expression of the virulence regulator *toxT* and the levels of the second messenger cyclic di-GMP (18, 52, 53). *V. cholerae* has a high tolerance for carbonates, and carbonates also inhibit the growth of *E. coli* at alkaline pH (43). We demonstrate here that high pH and carbonates have a synergistic effect at inhibiting *E. coli* growth (Fig. 6). We postulate that mutations in the citrate fermentation pathway reduce carbonate production enough to abrogate enteric growth inhibition.

Chakrabarty et al. (28) first observed that *V. cholerae* inhibits the growth of *E. coli* when grown on citrate-containing media, but we have demonstrated that external citrate is not required for this behavior (Fig. 1). This result is consistent with previous findings, where *V.*

*cholerae* was able to inhibit enteric growth on acetate and lactate medium (29). It is surprising that the *V. cholerae* C6706 Δ*citE* and Δ*citF* mutants were able to grow on Simmons' citrate agar (Table 1). Despite its widespread prevalence in *Vibrio* spp., little is known about *V. cholerae* citrate metabolism (15). In these mutant strains, citrate could be converted to oxaloacetate via the glyoxylate shunt, feeding into gluconeogenesis for the production of biomass. This pathway is mediated by isocitrate lyase (ICL), which *V. cholerae* C6706 carries (VC0736). Consistent with this alternate citrate metabolic pathway, we simulated *V. cholerae* metabolism using a previously constructed genome-scale metabolic model based on the V52 strain (54) with the software OptFlux (55). We found that disruption of ACLY did not impact *V. cholerae* growth *in silico* when citrate was the sole carbon source, with both producing equal biomass values (0.56). This process was dependent on the glyoxylate shunt, as disruption of both ACLY and ICL resulted in no biomass production *in silico*. Conversely, several of the *V. cholerae* strains examined in this study were not capable of citrate fermentation on Simmons' agar. Currently, there are not annotated genomic sequences for all of these *Vibrio* strains, but at least one strain (MZO-2) carries citrate lyase genes with >99% protein identity to the N16961 strain, and yet, it does not grow on Simmons' agar, suggesting that there may be differential expression of citrate metabolism genes among these strains. There are also examples of *V. cholerae* acquiring mutations in conserved metabolic pathways (56).

Citrate metabolism is not necessarily indicative that a *V. cholerae* strain inhibits enteric growth, suggesting other systems work in conjunction with citrate metabolism to create a hostile growth environment for *E. coli* and *S. flexneri* (Table 1). While we demonstrate that the citrate metabolic axis is required for *V. cholerae* C6706 inhibition of enteric growth, it is likely that other secreted molecules also contribute to this process, as we did not observe a consistent correlation among *V. cholerae* strains between growth on citrate, the pH of LB media after 48 h, and enteric growth inhibition (Table 1). This finding suggests that there are multiple ways in which different *Vibrio* strains can inhibit enteric growth. This hypothesis is similar to those made in previous studies, where they observe different patterns of *V. cholerae* enteric growth inhibition (28, 30). Notably, C6706 and N16961 are genetically similar (57, 58) and yet produce different pH after extended growth (Table 1); there is evidence that lab domestication has impacted some phenotypes of these strains, which could include enteric inhibition (59, 60). There is diversity among *Vibrio* strains in their metabolic pathways; for example, some *V. cholerae* El Tor biotypes have developed neutral fermentation pathways, resulting in 2,3-Butanediol production, to avoid creating potentially harmful organic acids (25, 26). It is possible that certain *V. cholerae* strains produce other secondary metabolites that also contribute to enteric growth inhibition, independent of citrate fermentation. One possibility is the production of polyamines, such as cadaverine which some *V. cholerae* species produce in high abundance (61, 62). The synthesis of cadaverine consumes protons and protects *V. cholerae* from acid stress, and also inhibits enteric bacteria at high pH (63). Further studies will reveal how other secondary metabolites contribute to *V. cholerae* inhibition of enteric bacterial growth.

This report describes an example of how *V. cholerae* C6706 uses metabolic products to outcompete other bacteria *in vitro*. Specifically, we propose that creating a highly alkaline environment is a mechanism for *V. cholerae* C6706 to generate a niche. The concept of *V. cholerae* using secreted metabolites to protect its niche is not entirely novel, as a previous study has shown that *V. cholerae* releases ammonium when grown on chitin to inhibit protist grazing (64). Creating a niche diminished of other species of bacteria would allow *V. cholerae* to suppress competing commensal bacteria and control its nutrient pool. In combination with other known systems, such as the T6SS, this mechanism of competition likely contributes to the survivability, adaptability, and success of *V. cholerae* as a pathogen and prominent aquatic bacterium.

## MATERIALS AND METHODS

**Strains and culture conditions.** Strains included in this study are *E. coli* MG1655 and *S. flexneri* 2457T (CFS100 [65]), as well as *V. cholerae* strains listed in Table S1 in the supplemental material. *V. cholerae* and *E. coli* strains were grown in lysogeny broth (LB) (1% tryptone, 0.5% yeast extract, and 0.5%

NaCl) at 37°C with shaking. As necessary, bacteria were grown in the presence of 50 $\mu$g/mL kanamycin, 100 $\mu$g/mL streptomycin, or 50 $\mu$ g/mL rifampicin. Cross streaking and conditioned medium assays were performed using LB or citrate media broth (CB). Briefly, nutrient broth no. 2 (Oxoid) was supplemented with NH$_4$Cl (0.03%), K$_2$HPO$_4$ (0.5%), and sodium citrate (0.5%) and includes EGTA (3.0 mg/mL).

To determine whether a bacterium could utilize citrate as a sole carbon source, strains were grown on Simmons' citrate agar. Ammonium dihydrogen phosphate (0.2 g/L), disodium ammonium phosphate (0.8 g/L), magnesium sulfate heptahydrate (0.2 g/L), and sodium chloride (5.0 g/L) salts were added to a solution of 2.0 g/L trisodium citrate, with 1.5% (wt/vol) agar.

**Cross-streaking assay.** The cross-streaking assay was performed as described previously (66). Briefly, liquid from an overnight culture of the producing bacteria was streaked down the center of an agar plate (LB or CB) using a sterile cotton-tipped stick, resulting in a ~3/4" wide stripe and was left to grow for 48 h at 37°C followed by 6 h at 4°C. The resulting growth was manually removed by scraping the plate with a sterile cotton-tipped stick, and the remaining bacteria were killed by exposure to chloroform vapor for 30 min. Residual chloroform was allowed to evaporate from the plate for 30 min in the fume hood. Afterward, the indicator strain (for example, *E. coli* or *S. flexneri* CFS100) was streaked in a line perpendicular to the producing bacteria from an overnight culture in the same manner. The plate was then incubated for 18 h at 37°C. For visualizing pH changes in the agar plate, an LB agar plate with 0.0032% (wt/vol) thymol blue was inoculated with a single streak with *V. cholerae* C6706 from an overnight culture and incubated statically for 24 h at 37°C.

**Conditioned medium assay.** The concentration of our LB medium prior to bacterial growth was approximately 7.0. Both *V. cholerae* and *E. coli* or *S. flexneri* CFS100 were grown for 48 h at 37°C. The resulting culture was centrifuged, the supernatant was retained, and the supernatant was filter sterilized using 0.22-$\mu$m Millipore polyvinylidene difluoride (PVDF) filters. The *V. cholerae* supernatant (either diluted with saline or not) or medium at a specific pH was mixed 1:1 with a 1:1,000 dilution of an *S. flexneri* CFS100 overnight culture in LB to a total volume of 100 $\mu$L in each well of a 96-well plate. The *V. cholerae* supernatant produced in LB was used to perform the inhibition assays, using supernatant dilutions ranging from 50% to 0% (final concentration). Plates were then incubated at 37°C, shaking at 200 rpm. OD$_{595}$ was measured at the 6-h time point using an Opsys MR plate reader.

LB was adjusted to various pHs with NaOH and autoclaved. Supernatant from *V. cholerae* grown in neutral LB was adjusted to multiple pHs with 1 M NaOH or 1 M HCl, as appropriate, and was filter sterilized using an 0.22-$\mu$m polyethersulfone (PES) membrane filter. The *V. cholerae* supernatant and LB at close pHs were combined 1:1, and their pHs were redetermined. The resulting media (either LB or 50% *V. cholerae* supernatant) were then used to perform growth inhibition assays as described above. Media were mixed 20:1 with a 1:100 inoculum of a *S. flexneri* CFS100 overnight culture. OD$_{595}$ was measured at the 6-h time point.

To determine if supernatants contained an inhibitory protein, sterile supernatants were prepared as described previously, and then these supernatants were either filtered through a 1-kDa nominal molecular weight (NMW) ultrafiltration disc (Millipore) or treated with proteinase K. Briefly, proteinase K powder was dissolved at a concentration of 20 mg/mL in sterile 50 mM Tris (pH 8.0) and 1.5 mM calcium acetate (cite proteinase K recipe). A total of 20 $\mu$L of this solution was added to 2 mL of supernatant and incubated at 50°C for 20 min. The supernatant was then incubated at 70°C for 10 min to inactivate the proteinase K, before it was used in our conditioned medium assay.

Methanol extraction of metabolites was performed as described previously, and extractions were performed in triplicate (67). Sterile supernatants were prepared as previously described above and then 600 $\mu$L cold MeOH was added to the 100 $\mu$L *V. cholerae* supernatant. The mixture was vortexed and then centrifuged for 2 min at maximum (max) speed at 4°C. A total of 100 $\mu$L chloroform was then added to the mixture, then vortexed, and centrifuged for 2 min at max speed at 4°C. A total of 300 $\mu$L water was then added to the mixture, which was then vortexed and centrifuged for 2 min at max speed at 4°C. The aqueous phase was then transferred to a new tube, and 300 $\mu$L MeOH was added. The mixture was evaporated by vacuum centrifugation, and the remaining material was resuspended in 100 $\mu$L sterile saline.

**Growth assays.** For growth curves of *S. flexneri* in different concentrations of the *V. cholerae* C6706 supernatant, overnight cultures of *S. flexneri* CFS100 were diluted (1:100) into 96-well plates containing LB with decreasing concentrations of the cell-free supernatant; plates were incubated at 37°C with shaking, and the optical density at 600 nm of bacterial cultures was measured and recorded every 20 min for 10 h using a BioTek Synergy H1 plate reader. Data were analyzed using the R package grofit (68), with the Richards growth model.

For growth curves of *S. flexneri* in different pH media, overnight cultures of *S. flexneri* CFS100 were diluted (1:100) into LB at different pHs, and the optical density at 600 nm of bacterial cultures was measured and recorded every hour for 7 h. For comparing the relative growth rates of *E. coli* and *V. cholerae*, overnight cultures of *V. cholerae* or *E. coli* were diluted (1:100) into CB in the presence or absence of bicarbonate buffered to pH of 7 or 9. OD$_{600}$ readings were recorded every hour. The resulting ODs were plotted on a graph versus time, and the portion of the graph that was linear was used to calculate a slope. The slope of *E. coli* was then divided by the slope of *V. cholerae* grown under the same growth conditions.

***In silico* analysis of *V. cholerae* metabolism.** *In silico* metabolism simulations were performed using a previously published *V. cholerae* genome-scale metabolic network reconstruction (54) and OptFlux (55). Although *V. cholerae* carries a sodium/citrate symporter (VC0795), there was no reaction for citrate transport in this model (54), so citrate exchange was modified to be cytoplasmic. We removed external boundary metabolites and used the core biomass production as our objective function and the parsimonious Flux-Balance Analysis simulation method. To simulate growth in Simmons' media, default environmental conditions were used, except that the lower bound of glucose exchange was set to 0 and the lower bound of citrate exchange was set to −20. Total biomass production was limited by carbon availability under these simulated conditions.

**Statistical analysis.** All statistical analysis was performed using GraphPad Prism. For supernatant inhibition assays, data were regressed using a nonlinear four-parameter variable slope inhibitor model, and $IC_{50}$ values were compared using a sum-of-squares F-test ($P < 0.05$). For growth analysis, data were compared using analysis of variance (ANOVA) ($P < 0.05$).

**Data availability.** All data generated or analyzed during this study are included in this published article and in File S1 in the supplemental material.

## SUPPLEMENTAL MATERIAL

Supplemental material is available online only.

**SUPPLEMENTAL FILE 1**, PDF file, 0.1 MB.

## ACKNOWLEDGMENTS

We thank R.A. Finkelstein for his thoughtful contributions to this study.

Work in the laboratory of S.P. is supported by the National Institutes of Health (NIH) R01 grant AI139103-01A1 and the Canadian Institutes of Health Research Operating Grants MOP-84473, MOP-137106. B.K. was a recipient of an NSERC PGS-D. Work in the laboratory of S.M.P. is supported by the NIH R37 grant AI016935-35. Work in the laboratory of B.J.K. is supported by the NIH R03 grant AI156432-01A1 and by startup funds and a grant from the Faculty Research and Creative Activities Award, Western Michigan University.

S.M.P., S.P., and B.J.K. contributed to conceptualization, data curation, funding acquisition, and supervision. B.K., M.E.B., J.J.B., C.N.C., and B.J.K. contributed to investigation, methodology, validation, and formal analysis. B.K. and B.J.K. contributed to writing, and all authors reviewed and edited the manuscript.

We declare no competing interests.

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
