## [Reviewer comments · Microbiology Spectrum]

Microbiology Spectrum

Vibrio cholerae* alkalizes its environment via citrate metabolism to inhibit enteric growth *in vitro

Benjamin Kostiuk, Mark Becker, Candice Churaman, Joshua Black, Shelley Payne, Stefan Pukatzki, and Benjamin Koestler

Corresponding Author(s): Benjamin Koestler, Western Michigan University

Review Timeline:

Submission Date:	November 29, 2022
Editorial Decision:	January 4, 2023
Revision Received:	January 23, 2023
Accepted:	February 22, 2023

Editor: Ana Weil

Reviewer(s): Disclosure of reviewer identity is with reference to reviewer comments included in decision letter(s). The following individuals involved in review of your submission have agreed to reveal their identity: Yang Fu (Reviewer #1)

Transaction Report:

DOI: <https://doi.org/10.1128/spectrum.04917-22>

January 4, 2023

Prof. Benjamin J Koestler
Western Michigan University
Department of Biological Sciences
1903 West Michigan Avenue
MS 5410
Kalamazoo, MI 49008

Re: Spectrum04917-22 (*Vibrio cholerae* alkalizes its environment via citrate metabolism to inhibit enteric growth *in vitro*)

Dear Prof. Benjamin J Koestler:

Thank you for submitting your manuscript to Microbiology Spectrum. We feel that this manuscript requires revisions to be published; please see reviewer comments. When submitting the revised version of your paper, please provide (1) point-by-point responses to the issues raised by the reviewers as file type "Response to Reviewers," not in your cover letter, and (2) a PDF file that indicates the changes from the original submission (by highlighting or underlining the changes) as file type "Marked Up Manuscript - For Review Only". Please use this link to submit your revised manuscript - we strongly recommend that you submit your paper within the next 60 days or reach out to me. Detailed instructions on submitting your revised paper are below.

Link Not Available

Sincerely,

Ana Weil

Journals Department
Reviewer comments:

Reviewer #1 (Comments for the Author):

Kostiuk et al found that *Vibrio cholerae* could metabolize citrate as a carbon source and turn the medium into an alkaline environment, which is harmful to the growth of other enteric bacteria. This presents a novel strategy for *V. cholerae* to antagonize the microbes in its surrounding. Overall, the methods and approach are detailed and solid. The experiments were performed clearly and are convincing.

Some minor suggestions may be better explained and improved.

1. The medium of long-term culture or growth on the agar plate for an extended time of *Vibrio cholerae* shows a significant inhibition for other enteric bacteria, which is actually artificially giving *V. cholerae* a first-mover advantage in growth. If the bacteria are co-cultured from the same concentration, can *V. cholerae* show similar inhibitory effects? For example, using a 0.22µm membrane/filter to separate the *V. cholerae* and other enteric bacteria, but their metabolites can still diffuse freely at the same time.

2. The in vitro experiments are done detailed and logically. Is it possible to detect the impact of *V. cholerae* on in vivo co-colonized enteric bacteria, and whether it can significantly reduce the CFU of *E. coli* or other enteric bacteria during mixed infection?[1] (After excluding T6SS contributions).

3. line 176: Transposon-based mutant is a wonderful strategy for quick screening, however, it could be better to use clean knockout strains for verification.

1. Liu, M., et al., CitAB Two-Component System-Regulated Citrate Utilization Contributes to *Vibrio cholerae* Competitiveness with the Gut Microbiota. *Infect Immun*, 2019. 87(3).

Reviewer #2 (Comments for the Author):

Microbes must compete with one another in infection and in environmental settings. Bacteria including those studied here have multiple mechanisms to aid in competition. Kostiuk and colleagues here study the important bacterium *Vibrio cholerae*, which can cause human disease cholera and is common in marine reservoirs. Prior work in the 1970s by Vora and others described the ability of *V. cholerae* to inhibit growth of intestinal/enteric bacteria including *E. coli* when cultured in citrate, using a simple cross streaking method and culture supernatants alone. Here the authors provide further insights into this phenotype. Evidence is provided that this inhibition by some strains of *V. cholerae* is due to elevation of pH, which is enhanced by citrate metabolism and hinders *E. coli* and *Shigella* but not *V. cholerae*. These results contribute to our understanding of how some *V. cholerae* strains may compete by modifying external conditions to favor their growth over others.

Major comments:

Lines 211-225. The pH values reported in Table 1 vary greatly. What is "significant elevated" pH? The results show only C6706 (7.7-9.5) and DL4211 (8.9-9.3) have pH 9.0 or higher but can you say the pHs achieved by the five strains are different from one another? Statistically? It is interesting that you observe differences between C6706 and N16961 which have few genetic differences. Can these few differences (other than QS) give you insights into the mechanism?

Minor comments:

Line 111. Abbreviations of medium typically lack periods. I suggest "C.B." be replaced by "CB" throughout. See next comment.

Line 122. despite being described as Lysogeny Broth in the Methods, it should also be described as such here. "L.B." should be abbreviated "LB".

Line 160-1. It is unclear what this means, why it is indicated here and how it connect to the question being asked. Where are the results showing this? Growth in conditioned medium has not been explained or introduced.

Line 169. "homologous genes". Homologous to what?

Line 188-9. Reference needed.

Line 211. The pH section beginning here should be the beginning of a new paragraph as this is an important turning point in the results.

Lines 246-250. Are the Thymol Blue results consistent with the pH changes shown in Table 1? Do C6706 citE, citF and oadA mutants not produce a blue zone? DL4211 yes? N16961 and O395 no?

Lines 310-312. References needed.

Line 173 and Fig 3C. It would be useful for the reader to include in Fig. 3C the chemical formulas of carbonic acid (H₂CO₃), bicarbonate (HCO₃), and carbonate (CO₃). This connects helps show that the CO₂ generated from citrate metabolism is converted into these molecules. It would also be helpful to refer in the text to each of the panels of Fig. 3.

Line 312. "A.K.I." should read "AKI"

Lines 370-1. "non-standard" metabolic pathway. Was this really shown? What does "non-standard" mean? What is standard? If this is referring to the glyoxylate shunt predicted by the in silico analysis, then a vc0736 mutant should be tested. As presented here though, the authors do not provide sufficient evidence to support this statement.

It would be beneficial in the text, perhaps the Discussion, to include a frank statement acknowledging that LB conditions do not mimic conditions found in a host or the environment.

Staff Comments:

Preparing Revision Guidelines

Please return the manuscript within 60 days; if you cannot complete the modification within this time period, please contact me. If you do not wish to modify the manuscript and prefer to submit it to another journal, please notify me of your decision immediately so that the manuscript may be formally withdrawn from consideration by Microbiology Spectrum.

Reviewer #1 (Comments for the Authors):

Kostiuk et al found that *Vibrio cholerae* could metabolize citrate as a carbon source and turn the medium into an alkaline environment, which is harmful to the growth of other enteric bacteria. This presents a novel strategy for *V. cholerae* to antagonize the microbes in its surrounding. Overall, the methods and approach are detailed and solid. The experiments were performed clearly and are convincing.

Some minor suggestions may be better explained and improved.

1. The medium of long-term culture or growth on the agar plate for an extended time of *Vibrio cholerae* shows a significant inhibition for other enteric bacteria, which is actually artificially giving *V. cholerae* a first-mover advantage in growth. If the bacteria are co-cultured from the same concentration, can *V. cholerae* show similar inhibitory effects? For example, using a 0.22um membrane/filter to separate the *V.cholerae* and other enteric bacteria, but their metabolites can still diffuse freely at the same time.

2. The *in vitro* experiments are done detailed and logically. Is it possible to detect the impact of *V.cholerae* on *in vivo* co-colonized enteric bacteria, and whether it can significantly reduce the CFU of *E.coli* or other enteric bacteria during mixed infection?[1] (After excluding T6SS contributions).

3. line 176: Transposon-based mutant is a wonderful strategy for quick screening, however, it could be better to use clean knockout strains for verification.

1. Liu, M., et al., *CitAB Two-Component System-Regulated Citrate Utilization Contributes to Vibrio cholerae Competitiveness with the Gut Microbiota*. Infect Immun, 2019. **87**(3).

Reviewer #1 (Comments for the Author):

Kostiuk et al found that *Vibrio cholerae* could metabolize citrate as a carbon source and turn the medium into an alkaline environment, which is harmful to the growth of other enteric bacteria. This presents a novel strategy for *V. cholerae* to antagonize the microbes in its surrounding. Overall, the methods and approach are detailed and solid. The experiments were performed clearly and are convincing.

Some minor suggestions may be better explained and improved.

1. The medium of long-term culture or growth on the agar plate for an extended time of *Vibrio cholerae* shows a significant inhibition for other enteric bacteria, which is actually artificially giving *V. cholerae* a first-mover advantage in growth. If the bacteria are co-cultured from the same concentration, can *V. cholerae* show similar inhibitory effects? For example, using a 0.22µm membrane/filter to separate the *V. cholerae* and other enteric bacteria, but their metabolites can still diffuse freely at the same time.

- Thank you for these comments, we agree that this is a very artificial setup, which we have modified to state more clearly in the Discussion (lines 282-287). While it would be interesting to examine if enteric inhibition occurs during co-culture, we feel that the proposed experiment would be challenging to interpret, given our current gaps in understanding of this phenomenon. *V. cholerae* grows faster than *E. coli* and *Shigella*, and so it would be difficult to parse effects due to nutrient depletion. We believe there are also other unknown factors that contribute to enteric inhibition (as discussed, line 354-357). We hope that our future studies will illuminate what these are, which would provide us with the genetic controls we would need for this experiment.

2. The in vitro experiments are done detailed and logically. Is it possible to detect the impact of *V. cholerae* on in vivo co-colonized enteric bacteria, and whether it can significantly reduce the CFU of *E. coli* or other enteric bacteria during mixed infection?[1] (After excluding T6SS contributions).

-Similar to above, we think it may be possible, but there would be confounding variables that would make this experiment difficult. We feel that discovering these other contributing factors is necessary to interpret the results of an in vivo experiment. We have modified our manuscript to include this as a limitation of our current study (lines 282-287).

3. line 176: Transposon-based mutant is a wonderful strategy for quick screening, however, it could be better to use clean knockout strains for verification.

-We agree that clean knockout strains coupled with complementation would be a more robust approach. Because we observed effects using these transposon mutants, we thought it was worthwhile including them in this study. We note that these are transposon mutants in the main text (line 171-173), so readers are aware of this limitation.

1. Liu, M., et al., CitAB Two-Component System-Regulated Citrate Utilization Contributes to *Vibrio cholerae* Competitiveness with the Gut Microbiota. *Infect Immun*, 2019. 87(3).

-Thank you for identifying this important study; we include discussion of this manuscript (lines 75, 166, 304, 323) as well as a citation for it (reference 15). We believe that our work here

supports this study, and potentially offers an additional mechanism by which citrate metabolism contributes to *V. cholerae* competitiveness in vivo, by modulation of environmental pH.

Reviewer #2 (Comments for the Author):

Microbes must compete with one another in infection and in environmental settings. Bacteria including those studied here have multiple mechanisms to aid in competition. Kostiuk and colleagues here study the important bacterium *Vibrio cholerae*, which can cause human disease cholera and is common in marine reservoirs. Prior work in the 1970s by Vora and others described the ability of *V. cholerae* to inhibit growth of intestinal/enteric bacteria including *E. coli* when cultured in citrate, using a simple cross streaking method and culture supernatants alone. Here the authors provide further insights into this phenotype. Evidence is provided that this inhibition by some strains of *V. cholerae* is due to elevation of pH, which is enhanced by citrate metabolism and hinders *E. coli* and *Shigella* but not *V. cholerae*. These results contribute to our understanding of how some *V. cholerae* strains may compete by modifying external conditions to favor their growth over others.

Major comments:

Lines 211-225. The pH values reported in Table 1 vary greatly. What is "significant elevated" pH? The results show only C6706 (7.7-9.5) and DL4211 (8.9-9.3) have pH 9.0 or higher but can you say the pHs achieved by the five strains are different from one another? Statistically? It is interesting that you observe differences between C6706 and N16961 which have few genetic differences. Can these few differences (other than QS) give you insights into the mechanism?

-Thank you, this is a great question! We observed a lot of variance in the pH readings, which we found interesting. There was one outlier which was confounding statistical analysis; in this revision, we added replication for the C6706 and citF mutants, and included statistical analysis in Table 1. We have removed this outlier, and note this in the Table 1 legend (lines 761-764), and have modified the text (now Lines 222-224) to note this variance. As to the nature of this variance, because we are creating conditions that may select for spontaneous compensatory mutations (prolonged incubation, leading to starvation conditions), we suspect that this may be driving the variance we observe. However, this is very speculative, hence we refrain from discussing this in the manuscript.

We also found it interesting that there was a difference between C6706 and N16961. We examined the role of HapR, which is known to be different between the two strains, but a C6706 hapR mutant had no defect in enteric inhibition (Fig. 4B). Melanie Blokesch has done some really nice work demonstrating effects of lab domestication on *V. cholerae* C6706 (<https://dx.doi.org/10.1128/mSphere.00098-16>, <https://doi.org/10.1111/1462-2920.15214>), where she observed variation in QS amongst the same strain from different labs; thus it is possible other mutations could have also occurred between our C6706 and N16961. We agree this could be an excellent way to identify mechanisms contributing to the pH dependent part of this process, but would require additional genome sequencing. We now note this in our discussion (lines 360-363).

Minor comments:

Line 111. Abbreviations of medium typically lack periods. I suggest "C.B." be replaced by "CB" throughout. See next comment.

-Corrected

Line 122. despite being described as Lysogeny Broth in the Methods, it should also be described as such here. "L.B." should be abbreviated "LB".

-Corrected

Line 160-1. It is unclear what this means, why it is indicated here and how it connects to the question being asked. Where are the results showing this? Growth in conditioned medium has not been explained or introduced.

-We agree this sentence was confusing, and raised an inappropriate comparison between our study and the cited one, thus we have deleted it.

Line 169. "homologous genes". Homologous to what?

-we have deleted the word homologous.

Line 188-9. Reference needed.

-We have added additional references (references 38 and 39)

Line 211. The pH section beginning here should be the beginning of a new paragraph as this is an important turning point in the results.

-Corrected.

Lines 246-250. Are the Thymol Blue results consistent with the pH changes shown in Table 1? Do C6706 citE, citF and oadA mutants not produce a blue zone? DL4211 yes? N16961 and O395 no?

-We have modified the text in Line 249 to indicate this is consistent with the pH change we observed in Table 1. We only performed this analysis on C6706, so we cannot state if it is consistent with other strains.

Lines 310-312. References needed.

-Reference for human bicarbonate secretion have been added (reference 50).

Line 173 and Fig 3C. It would be useful for the reader to include in Fig. 3C the chemical formulas of carbonic acid (H_2CO_3), bicarbonate (HCO_3^-), and carbonate (CO_3^{2-}). This connects and helps show that the CO_2 generated from citrate metabolism is converted into these molecules. It would also be helpful to refer in the text to each of the panels of Fig. 3.

-Thank you for this suggestion, we have added this text in Fig. 3C.

Line 312. "A.K.I." should read "AKI"

-Corrected

Lines 370-1. "non-standard" metabolic pathway. Was this really shown? What does "non-standard" mean? What is standard? If this is referring to the glyoxylate shunt predicted by the in silico analysis, then a vc0736 mutant should be tested. As presented here though, the authors do not provide sufficient evidence to support this statement.

-We agree that this is poor wording. We have changed the sentence (now line 375) to "This report describes an example of how *V. cholerae* C6706 uses metabolic products to outcompete other bacteria in vitro."

It would be beneficial in the text, perhaps the Discussion, to include a frank statement acknowledging that LB conditions do not mimic conditions found in a host or the environment.

-Thank you for this suggestion, along with limitations identified by reviewer 1, we have added this statement (lines 282-287).

February 22, 2023

Prof. Benjamin J Koestler
Western Michigan University
Department of Biological Sciences
1903 West Michigan Avenue
MS 5410
Kalamazoo, MI 49008

Re: Spectrum04917-22R1 (*Vibrio cholerae* alkalizes its environment via citrate metabolism to inhibit enteric growth *in vitro*)

Dear Prof. Benjamin J Koestler:

Thank you for this interesting manuscript!

Your manuscript has been accepted, and I am forwarding it to the ASM Journals Department for publication. You will be notified when your proofs are ready to be viewed.

Sincerely,

Ana Weil
Editor, Microbiology Spectrum
